# Association of Vitamin D Deficiency and Insufficiency with Pathology in Hospitalized Patients

**DOI:** 10.3390/diagnostics13050998

**Published:** 2023-03-06

**Authors:** Sandica Bucurica, Ioana Prodan, Mihaela Pavalean, Corina Taubner, Ana Bucurica, Calin Socol, Roxana Calin, Florentina Ionita-Radu, Mariana Jinga

**Affiliations:** 1Department of Gastroenterology, “Carol Davila” University of Medicine and Pharmacy Bucharest, 020021 Bucharest, Romania; 2Department of Gastroenterology, “Carol Davila” University Central Emergency Military Hospital, 010825 Bucharest, Romania; 3Medical Analysis Laboratory, “Carol Davila” University Central Emergency Military Hospital, 010825 Bucharest, Romania; 4Department of Gastroenterology, Pucioasa City Hospital, 135400 Pucioasa, Romania

**Keywords:** vitamin D deficiency, hospitalized, 25(OH)D, Romania

## Abstract

Vitamin D deficiency is one of the most common medical conditions, with approximately one billion people having low vitamin D levels. Vitamin D is associated with a pleiotropic effect (immunomodulatory, anti-inflammatory and antiviral), which can be essential for a better immune response. The aim of this research was to evaluate the prevalence of vitamin D deficiency/insufficiency in hospitalized patients focusing on demographic parameters as well as assessing the possibility of its associations with different comorbidities. Of 11,182 Romanian patients evaluated in the study over 2 years, 28.83% had vitamin D deficiency, 32.11% insufficiency and 39.05% had optimal vitamin D levels. The vitamin D deficiency was associated with cardiovascular disorders, malignancies, dysmetabolic disorders and SARS-CoV2 infection, older age and the male sex. Vitamin D deficiency was prevalent and showed pathology association, while insufficiency of vitamin D (20–30 ng/mL) had lower statistical relevance and represents a grey zone in vitamin D status. Guidelines and recommendations are necessary for homogeneity of the monitoring and management of inadequately vitamin D status in the risk categories.

## 1. Introduction

Vitamin D deficiency is one of the most common medical condition, with approximately one billion people having low vitamin D levels [1]. Vitamin D is a fat soluble vitamin synthetized from precursors as 7-dehydrocholesterol into vitamin D3 (cholecalciferol) which represents the main source and from plant ergosterol into vitamin D2, through a multistep process (25-hydroxylation, 1α-hydroxylation, and 24-hydroxylation) to the biologic active form 1,25-dihydroxvitamin D_3_ [1,25(OH)_2_D_3_], respective 1,25(OH)_2_D_2_ [2,3]. The serum concentration of 25(OH)D, as an intermediate metabolite is the serum biomarker of vitamin D status, as it represents the main storage form and it is reliable to be measured, because the 1,25(OH)2 D form has around 4 h half-life, while 25(OH)D reaches 3 weeks [4].

Vitamin D production is secondary to conversion of the provitamin D3 into the skin by sunlight, Ultraviolet B rays (UVB) [1].

The levels of serum 25 hydroxyvitamin D (25(OH) vitamin D) are influenced by uptake, season, latitude, gender, ethnicity, environmental factors, sunscreen, skin pigmentation, age, intestinal absorption and use of supplements [1].

Vitamin D is essential for calcium homeostasis by increasing the absorption of the dietary calcium, reducing the calcium excretion and mobilizing calcium from the storages. There are ubiquitously distributed receptors for vitamin D in the human body with a series of roles, many still unfound. Vitamin D can be synthesized into the skin; however, its production can be decreased by a series of factors: clothing, sunscreens and reduced sunlight. The deficit of vitamin D is not only associated with bone pathology, but it is also related to cardiovascular diseases, diabetes, cancer, autoimmune disorders and infectious disorders [5].

Vitamin D has a pleiotropic effect (immune modulatory, anti-inflammatory and antiviral), which can be essential for a better immune response [5,6].

The conventional and well known pathway of vitamin D action involves the steroid vitamin D receptor (VDR) mediation and calcium absorption modulation with major implication in bone metabolism, but there are various non-conventional systemic effects of vitamin D [2].

In cancer, vitamin D and VDR regulation are involved in apoptosis, invasion, inhibition of inflammatory cytokines and regulation of microRNA, but there are equivocal results regarding relation of causality because the of the lack of standardization of 25(OH)D testing method and timing (before and after the diagnosis) [2,7,8,9,10].

Regarding the cardiovascular system and vitamin D relationship, it is stated that VDR are found in endothelial cells, in smooth muscle tissue and lack of vitamin D promotes atherogenesis [11]. In addition, VDR is expressed in pancreatic cells and mediates transcription in stellate cells, suggesting the mechanism of involvement in metabolic diseases such as diabetes mellitus [12,13].

There are still gaps in establishing thresholds for adequate levels of 25(OH)D and in the most recent statements and consensus, it is mentioned that the present data are not sufficient to define certain vitamin D status thresholds, because of the lack of standardization, but at least 20 ng/mL 25(OH)D should be achieved, treatment is recommended for values below, and may be considered for serum 25(OH)D < 30 ng/mL.

Moreover, the recommendations for dose supplementation or treatment of vitamin D inadequacy vary according to age, risk category, geographical situation and regional authority or medical society, creating a large heterogeneity in management of vitamin D deficit and leaving place for debates, but there is consistency in the agreement that general population screening for vitamin D deficiency is not recommended [14,15,16].

In 2019, Romanian Ministry of Health released a recommendation for vitamin D status assessment in children, pregnant women and adults. According to this, there are several risk categories of adult population that are exposed to vitamin D deficiency and measurement of 25(OH)D is recommended for adults with: chronic diseases, low ultraviolet light exposure, diseases of liver, renal diseases, maldigestion and malabsorption, cardio-vascular diseases, metabolic and nutritional impairment, cancer, allergies, autoimmune diseases, endocrinopathies and musculoskeletal diseases [17,18]. Moreover, the pandemic era started in March 2020 and serum 25(OH)D were assessed in the context of COVID-19 risk.

According to the Romanian Ministry of Health recommendation for adults, a value of 25(OH)D between 10–20 ng/mL (25–50 nmol/L) is considered deficient, while <10 ng/mL (<25 nmol/L) represents a severe deficiency.

Although a level >20 ng/mL (>50 nmol/L) may be considered sufficient, American Endocrine Society Guidelines suggest that a level >30 ng/mL is associated with lower risk of osteomalacia. A level between 20–30 ng/mL is considered insufficiency, while values of 25(OH)D > 250 nmol/L (100 ng/mL), are considered toxic [1,17]. Moreover, there are no recommendations of vitamin D status screening in overall healthy populations [18].

For inpatients, previous studies showed that vitamin D deficiency was associated with higher probability of exacerbation of the disease and need for hospitalization [19], with longer hospitalization period and worst outcome of hospital admission, and higher mortality rates in intensive care units [20,21,22]. In addition, residents of hospices and hospitalized patients presented the highest risk for deficient vitamin D compared with outpatients, and longer hospitalization of those with lower 25(OH)D [23], and overall hospitalized people tend to have higher rates of deficiency of 25(OH)D [24]. 

A particular attention was paid to COVID-19 and vitamin D beginning with the first wave of pandemic, since that multiple observational studies showed that low serum levels of 25(OH)D were associated with severe disease, but not showing a causal relationship. One of the most recent umbrella review of multiple meta-analyses over COVID-19 and vitamin D status reported a significant risk for acquiring the infection, for severity of progression and for high incidence of the mortality in cases with low serum of 25(OH)D, and it was suggested a potential benefit of vitamin D administrated after the diagnosis [25].

Although the etiopathogenetic mechanisms of rickets/osteomalacia is directly related to vitamin D deficiency, the literature is inconsistent regarding the cause and effect relationship between lower 25(OH)D and occurrence or progression of a specific disease or if lack of vitamin D could be a mark of deteriorated health condition.

The aim of the study was to evaluate the prevalence of vitamin D deficiency/insufficiency in hospitalised patients, considering demographic factors and to establish an association with different comorbidities.

## 2. Materials and Methods

Individuals who had their serum 25 hydroxyvitamin D (25(OH)D) measured at the Central Military Emergency University Hospital, Bucharest, Romania (latitude 44 N) over a 24-month period, starting 1 June 2020, until 31 May 2022, were included in this retrospective cross-sectional study.

Inclusion criteria: Patients >18 years old, admitted in Central Military Emergency University Hospital, Bucharest, Romania whom 25(OH)D level were measured in our laboratory and had a discharge diagnostic coded according to International Classification of Diseases, Tenth Revision, Clinical Modification (ICD-10-CM).

Exclusion criteria: patients under 18 years old, outpatients or patients that presented to the emergency room, but were not admitted to the hospital, patients with multiple admissions, pregnant women, patients with unspecific diagnosis (according to ICD-10-CM).

Gender, age, chronic diseases and 25(OH)D serum levels of the patients were evaluated. Adults were divided into five groups based on age (<20, 20–39, 40–59, 60–79, ≥80) (Figure 1).

The seasons were split into spring (March–May), summer (June–August), autumn (September–November) and winter (December–February).

In the last decade, the number of vitamin-D-related articles increased, with more concern for different afflictions. The goal was to focus on the largest, most relevant, and most recent studies.

Pathologies included were considered according to the first discharge diagnostics and were categorised according to International Classification of Diseases, Tenth Revision, Clinical Modification (ICD-10-CM). Additionally, there were used, as distinct codes, the new agreed codes for positive status of COVID-19 (since 1 March 2020); respective U07.1 and coronavirus infection B97.2, according to National Institute of Health Services Management (Institutul National de Management al Serviciilor de Sanatate). Additional to the primary discharge diagnostic, the secondary diagnostics which have been taken into consideration were grouped in five categories: cardiovascular disorders, dysmetabolic status (including obesity, diabetes and dyslipidemic disorders), endocrine disorders (excluding diabetes and dyslipidemic disorders), malignancy and COVID-19.

Pathologies/comorbidities with non-specific codes (Z00–Z99), (V01–X59) or special purposes codes (U00–U85) were excluded, except for U07.1 used for COVID-19, which was taken into account. The codes for pregnancy, childbirth and the puerperium (O00–O99), certain conditions originating in the perinatal period (P00–P96) and congenital malformations, deformations and chromosomal abnormalities (Q00–Q99) and multiple admissions were also excluded. In addition, we excluded patients under 18 years of age, outpatients or patients that presented to the emergency room, but were not admitted to the hospital.

Serum 25(OH)D was measured using chemiluminescence microparticle immunoassay (CMIA) for quantitative determination of 25-hydroxyvitamin D in human serum and plasma, with a measuring interval between 3.5 to 154.2 ng/mL (8.8 to 385.5 nmol/L) analyser. The same method of measuring was used for all the patients.

Serum vitamin D levels have been categorized as deficiency (25-OH vitamin D <20 ng/mL), insufficiency (20 to 30 ng/mL) and vitamin D optimal values ≥30 ng/mL [1].

The study was conducted in accordance with the Declaration of Helsinki and was approved by the Committee of Ethics from Central Emergency University Military Hospital no 572/13/01/2023. Informed consent was obtained for using and analysing data in scientific purposes from all subjects involved.

We focused on the largest studies and most recent ones published for our discussion chapter. We excluded the articles published in languages other than English.

### Statistical Analysis

Descriptive analyses were used for general characteristics and quantitative data were analysed using mean, median and standard deviation. Numbers and percentages were used for categorical values. Cross tabs and a chi-square test were used to examine the associations between categorical variables, including the adjusted residuals for the variation for the larger sample size. The test was considered valid when the test statistics in chi-squared distributed under the null hypothesis, specifically Pearson’s chi-squared test, Relative risk, odd ratio (OR) and 95% confidence intervals were calculated for association between categorical variables.

A *p*-value < 0.05 was considered significant. Adjusted residuals were considered with positive association >2 and negative association <−2 and for OR > 1 was considered statistically significant for the outcome occurring and <1 with lower odds of outcome. We tested for normal distribution the continuous variables in the whole group (11,182 subjects) using the Kolmogorov–Smirnov test for skewness and kurtosis and all continuous variables are expressed as mean ± standard deviation, median (min, max). The diagnostic accuracy was assessed with area under the curve (AUC) analysis of receiver operating characteristic (ROC) curves. Statistical analyses were performed using IBM SPSS^®^ version 26.

## 3. Results

### 3.1. Overall Demographic Variations

Data were collected from 14,090 patients who had their serum vitamin D level determined between 1 June 2020 and 31 May 2022 at the Central Military Hospital, Bucharest and were included in the analysis, according to inclusion criteria, 11,182 (65.9% women, *n* = 7366 and 34.1% males, *n* = 3186) with mean age 55.22 and median of 55 (18–102) (Table 1), but the age was not normally distributed, being skewed to the left (*p* < 0.01, skewness = −0.175, kurtosis = −0.603).

The prevalence of vitamin D insufficiency (20–30 ng/mL) was overall 32.1% (*n* = 3591)—from which 34.5% were males (*n* = 1239) and 65.5% were females (*n* = 2352) (*p* = 0.5634), while the overall vitamin D deficiency (<20 ng/mL) was 28.8% (*n* = 3224), 39.2% (*n* = 1265) males (*p* < 0.01; OR = 1.369; 95%Cl [1.257 to 1.490]) and 60.7% (*n* = 1959) females (*p* < 0.0001).

Overall, median levels of 25(OH)D were lower in the male group; 25.1 ng/mL < 27.2 ng/mL for females (Table 1).

Optimal vitamin D, ≥30 ng/mL, has been found in 39.05% (*n* = 4367) of the patients; 30.04% (*n* = 1312) males and 69.95% (*n* = 3055) females included in the study (*p* < 0.0001) (Table 2 and Appendix A, Figure 2).

The level of 25(OH)D was not normally distributed, being skewed to the right. The median level of 25(OH)D was 26.45 ng/mL (with range 3.5–154.2), with coefficient of skewness 1.384, coefficient of kurtosis of 6.283 with the Kolmogorov–Smirnov test for normality (*p* < 0.01).

### 3.2. Group Age Variations

In terms of age group, patients under 20 years of age were the fewest in this study, 0.82% (*n* = 92), 27.17% (*n* = 25) having insufficiency, 41.30% (*n* = 38) with deficiency and 31.52% (*n* = 29) with optimal levels of 25(OH)D.

The most prevalent age groups were 40–59 and 60–79 years of age. A total of 40.4% of patients (*n* = 4517) were in the 40–59 years of age group, 22.76% (*n* = 1028) of them presenting deficiency, 33.96% (*n* = 1534) insufficiency and 43.28% (*n* = 1955) with sufficient vitamin D.

For the age group 60–79 years, representing 36.6% (*n* = 4094) of the patients, 38.15% (*n* = 1562) had optimal levels of vitamin D, 29.46% (*n* = 1206) had insufficiency and 32.39% (*n* = 1326) had vitamin D deficiency. For the patients >80 years old (5.7% of all patients, *n* = 641), 26.05% (*n* = 167) had optimal values of vitamin D, 50.55% (*n* = 324) had vitamin D deficiency and 23.4% (*n* = 150) had insufficiency. Median 25(OH)D was lower in the extremities of age groups (Table 1 and Table 2, Figure 3).

### 3.3. Seasonal Variation

Based on the time period when 25(OH)D levels were tested, of all 11,182 patients, 20.8% had their 25(OH)D levels tested in winter, 31.1% of patients in spring, 25.0% in summer and 23.1% in the fall. The median level of the 25(OH)D for spring was 24.3 (3.5–154.2), for summer was 27.5 (3.5–154.2), for autumn was 28.7 (3.5–115.6) and for the winter season was 25.5 (3.5–154.2) (Table 1).

The vitamin D deficiency has been predominant in spring and winter. The vitamin D deficiency was 24.4% (*n* = 787) in winter, 16.6% (*n* = 534) in autumn, 20.6% (*n* = 665) in summer and 38.4% (*n* = 1238) in spring. Vitamin D insufficiency was predominant in the spring season (30.8%, *n* = 1105), summer (27.3%, *n* = 982) and autumn (24.2%, *n* = 868). Among patients with optimal 25(OH)D values, 26.1% (*n* = 1139) had 25(OH)D levels determined in the spring season, 26.3% (*n* = 1147) in the summer, 27.0% (*n* = 1180) autumn and 20.6% (*n* = 901) in winter.

A total of 24.5% of the study group (*n* = 2737) was evaluated in 2020, 42.2% (*n* = 1155) having normal 25(OH)D values, 22.8% (*n* = 625) having vitamin D deficiency and 34.9% (*n* = 957) having insufficiency. A total of 50.1% (*n* = 5606) of patients was evaluated in 2021, among whom 39.44% (*n* = 2211) with optimal values of 25(OH)D, 31.50% (*n* = 1766) with vitamin D deficiency and 29.05% (*n* = 1629) with insufficiency. For 2022, there was 25.4% (*n* = 2839) of the patients; 35.25% (*n* = 1001) with optimal values of 25(OH)D, 34.16% (*n* = 970) having vitamin D deficiency and 30.57% (*n* = 868) insufficiency.

### 3.4. Pathologies Variations 

Different pathologies were evaluated and an association with vitamin D status has been determined for some afflictions.

Vitamin D status (deficiency, insufficiency and optimal) in each pathology categorised as first discharge diagnostic was determined and represented in Table 3.

Regarding the association with one pathology, respecting the first discharge diagnostic, we found a significant association between vitamin D deficiency and infectious disease (*p* < 0.01, OR = 3.762, 95% confidence interval [CI] 2.584–5.475), malignant neoplasm (*p* < 0.01, OR = 2.673, 95% CI [1.859–3.845]), mental and behavioural disorders (*p* = 0.17, OR = 1.259, 95% CI [1.042–1.521]), other forms of heart disease and pulmonary heart diseases (*p* < 0.001, OR = 1.714, 95% CI [1.481–1.985]), diseases of the respiratory system (*p* < 0.001, OR = 1.26, 95% CI [1.124–1.413]), urinary and renal diseases (*p* < 0.001, OR = 2.358, 95% CI [1.801–3.088]) (Appendix A).

Because most of the patients had multiple pathologies, the 25(OH)D level has been evaluated also depending on the secondary diagnostic pathologies grouped in five categories: malignancies, cardiovascular disorders, dysmetabolic status, endocrine disorders and COVID-19. It has been shown that 41.9% of all included patients presented cardiovascular disorders; from them, a high percentage, 52.1%, had vitamin D deficiency (*p* < 0.0001) at a significantly higher rate compared to patients with cardiovascular disorders and levels >20 ng/mL of serum 25(OH)D.

The dysmetabolic status has been found in 38.4% of the patients and those also presented an increased percent of vitamin D deficiency (44.3%) compared to insufficient and optimal (38.6%, and, respectively, 34%). Although malignancy was found in only 3.8% (*n* = 420) of all patients studied, we found a high prevalence of vitamin D deficiency (45.7%; *n* = 190; *p* < 0.0001) among them. In the group of patients with endocrine disorders, deficiency and insufficiency had a prevalence equal to that of normal 25(OH)D levels (one third of patients falling into each category).

However, when we took into consideration the secondary diagnostics of the patients, we found an association of vitamin D deficiency with cardiovascular diseases (*p* < 0.0001; OR = 1.785; 95% Cl [1.257–1.490]), metabolic disorders (*p* < 0.01; OR = 1.409; 95% Cl [1.296–1.531]), COVID-19 (*p* < 0.0001; OR = 1.381; 95% Cl [1.244–1.534]) and with malignancy (*p* < 0.0001; OR = 2.11; 95% Cl [1.702–2.626]). The vitamin D deficiency is not associated with endocrine disorders (*p* = 0.7851). Moreover, the vitamin D insufficiency has been associated with cardiovascular diseases (*p* < 0.0001; OR = 0.848; 95% Cl [0.780–0.921]) and malignancy (*p* < 0.0001; OR = 0.727; 95% Cl [0.577–0.916]). There has not been found an association between vitamin D insufficiency and COVID-19 infection (*p* = 0.1938) and with the endocrine comorbidities (*p* = 0.9583) (Table 4, Table 5, Appendix A).

The results from the ROC curve analysis between vitamin D levels and renal impairment showed that the AUC was 0.621 (95% CI [0.579–0.662]; *p* < 0.001), showing acceptable accuracy of the diagnostic. The ROC curve analysis between 25(OH)D levels and infectious diseases showed that the AUC was 0.700, showing acceptable ability for discrimination of infective patients (95% CI [0.643–0.757]; *p* < 0.001). The results from the ROC curve analysis between 25(OH)D levels and renal impairment showed that the AUC was 0.621 (95% CI [0.579–0.662]; *p* < 0.001), showing acceptable accuracy of the diagnostic. The results from the ROC curve analysis between 25(OH)D levels and malignancy showed that the AUC was 0.632 (95% CI [0.578–0.687]; *p* < 0.001), revealing a moderate diagnostic accuracy.

## 4. Discussion

### 4.1. Overall Deficiencies and Insufficiencies and Demographic Variations

Vitamin D deficiency and insufficiency were observed in 60.9% (*n* = 6816) of 11,282 patients evaluated in the study, 28.8% having a deficit in vitamin D (*n* = 3224) and 32.1% having vitamin D insufficiency (*n* = 3592). A study conducted in Turkey (latitude 38 N) on 22,044 hospitalised patients that examined the deficiency and insufficiency of vitamin D has shown that 89.4% had low levels of 25(OH)D (78.6% females, 21.4% males) [26]. The prevalence of vitamin D deficiency is estimated at 5.9% in US [27] and 13% in Europe [28], while the prevalence of vitamin D insufficiency has been reported as 24% in US, and 40% in Europe [27,28,29,30]. It is clear that vitamin D deficiency represents a global health issue.

A Romanian study that evaluated the level of the 25(OH)D with a male to female ratio 1:2.9 emphasized the need for supplementation of vitamin D especially for older age, females and winter season [31]. In one of the most recent studies published in Romania, it was shown that females have lower levels of serum 25(OH)D compared with males [32].

In our study, we found that males have 1.36 times higher odds of having vitamin D deficiency and 1.02 times chances of insufficiency, while females presented a lower odds for both categories, although the females predominated in the sample (twice more numerous). We emphasise the fact that one third of all males presented <20 ng 25(OH)D serum level, while only 26% of all females presented deficient levels.

A previous study from Romania, performed on samples from 7544 patients, reported an overall mean of 25(OH)D of 27.20 ± 16.76, (34.45 ± 21.56 for males and 28.42 ± 14.45 for females), with an overall deficiency of 26%. There are limitations because there were also children included and the patients were referred from private practice or hospital, with no data about their diagnosis [31]. In contrast, in our study, males had lower values comparative to females. Similar to our results, there is another study from the Romanian population that reported lower mean values in males comparative with females for the 19–30 age group >70 years old, and one prospective study found male inpatients more deficient than females [33,34].

We should consider that in the last few years, vitamin D supplementation was used intensively, especially in the pandemic era and we have no data regarding vitamin D supplementation in our study population.

Moreover, French studies on healthy volunteers showed a dynamic of vitamin D status in population over the years, and showed a decreasing tendency of vitamin D deficiency prevalence from 57.7% to 34.6% in a 2 decades period, but with suboptimal level still prevalent, so updated data are necessary to have an accurate overview and to compare [35].

For Europe, there has been reported a prevalence of 53% of suboptimal vitamin D (severe deficiency <12 ng/mL prevalence of 13% and 40% between 12 ng/mL and 29 ng/mL) [16], but there are scarce data from the Eastern European area and individual studies showed a poorer vitamin D status (mean 25(OH)levels <20 ng/mL) compared with western and northern areas of the European continent [14].

### 4.2. Group Age Variations

In the age group distribution, it was revealed there was a predominance of the deficiency/insufficiency in middle-aged groups and, also, in the elderly, >80 years old, where half of the patients had vitamin D deficiency, similarly to the literature data.

The highest prevalence of vitamin D deficiency has been found in children [36], childbearing women [37] and elderly people (>75 years old) [38]. There has been a comparison made between China and the US, revealing different predictors for low levels of 25(OH)D. China individuals affected are of older age, females, people with high income, while for US, the most affected are males, lower income, no physical activity, overweight and obese [39]. Ethnic minorities and individuals with altered health are found in both countries [39].

Moreover, an increased incidence of vitamin D deficiency for older adults related to lower capacity for cutaneous synthetizations and less sunlight exposure has been postulated [40]. Schöttker has shown that there is a decrease of 3 nmol/L serum 25(OH)D for every 10 years of age in a large cohort (*n* = 9940) [41].

Romanian data from more than 8000 patients showed a high deficiency in elderly patients [32]. Our data are concordant with the literature data sustaining, once again, that people <20 or >60 years old (especially > 80) are at risk of having vitamin D deficiency rather than insufficiency (which is more encountered in 40–60 years age group), but another study from Romania on patients from private practice and hospital referral with indication for 25(OH)D serum measurement showed a variation of vitamin D deficiency slightly higher for different age groups [31].

In our study, we found a relatively similar prevalence of deficiency compared with the literature, slightly lower for the 20–39 age group, the 40–59 age group and the 60–79 age group, with mean levels of serum 25(OH)D, at the lower limit, especially in the 40–59 age group and 60–79 group and lower for the 20–39 age group and >80 years compared with previous published data [31,33].

We hypothesise that this could be the result of increasing awareness of vitamin deficiency correction and possible supplementation due to coronavirus pandemic.

### 4.3. Seasonal Variation

There was a slight predominance of the winter and spring season for vitamin D deficiency and for vitamin D insufficiency; it has been linked to spring and summer seasons. The sun exposure and seasonal dependency of 25(OH)D levels is well known and our study shows that in spring and winter seasons there is a higher risk of having vitamin D deficiency, while the summer season is related to the insufficiency. The months which are associated with higher rates vitamin D deficiency are from January until May, with no effect in June and with no relation to the others months.

Most studies showed a higher prevalence of the vitamin D deficit in autumn–winter seasons [28,42,43,44]. The median levels of 25(OH)D are high in summer and autumn seasons and are low in winter and spring. This aspect is correlated with the sun exposure. The most recent data from Romania revealed that the seasonal variability is descending from September to March (highest to lowest level of serum 25(OH)D), accordingly to the seasonal pattern in Romania, which has four seasons [32].

A Brazilian study assessed the vitamin D deficiency/insufficiency depending on seasons, showing higher prevalence of the deficiency at the end of autumn and winter (21.7% and 8.7%) than at the end of spring and summer (1.5% and 0%) for male individuals who have performed outdoor activities wearing shorts and t-shirts twice a week for 6 h. It has been shown there is a seasonal variation of the levels of 25(OH)D in obese patients following bariatric surgery. In 1071 evaluated patients, the highest levels have been assessed during summer (33.52 ± 12.70 ng/mL) and lowest levels during spring (29.72 ± 12.34 ng/mL and winter (29.90 ± 14.80 ng/mL) [42]. Moreover, the prevalence of the insufficiency, <30 ng/mL, was higher in the study group in autumn versus summer (88.4% vs 43.8%) [43].

A Mexican longitudinal study published in 2017 evaluated the season variation of vitamin D, showing lower levels during winter, increasing during spring with maximum levels during summer and autumn. The vitamin D deficiency was prevalent in winter, 60.0%, while, during summer, the deficiency of 25 (OH)D was 40.0%. The vitamin D insufficiency was 87.5% in the winter–autumn period and 91.3% during spring–summer [44]. A study from 2016, evaluating 55,844 European individuals, showed that 17.7% had low levels of 25(OH)D, <30 nmol/L in the October–March period and 8.3% during the April–November period. Moreover, the incidence of levels of 25(OH)D < 50 nmol/L, according to different definition, was 40.4% [28].

We found similar seasonal variation with results from another Romanian study, with the lowest serum 25(OH)D in the spring and the highest in the fall season, consistent with global seasonal variation of 25(OH)D levels [32].

### 4.4. Pathologies Variations

Infectious disease are under the influence of vitamin D activity as an independent immunomodulation factor that contributes to immune defence through pleiotropic action [45]. Overall, the vitamin D deficiency is considered as a predisposing factor for acquiring an infection and in our study, we found that patients with infectious diseases were predisposed 3.7 times more to have lower than 20 ng/mL serum 25(OH)D.

The relationship between COVID-19 infection and vitamin D has been studied intensively in the last period [46,47]. 25(OH)D regulates the renin angiotensin system and ACE2 expression [46]. It was postulated low levels of 25(OH)D influenced susceptibility, severity and mortality of COVID-19 infection [48,49,50]. Vitamin D supplementation in COVID-19 was evaluated in some cohorts, revealing anti-inflammatory, antiviral, apoptotic and autophagic activity through antimicrobial peptides [47,48,49,50,51]. Vitamin D deficiency was associated, in our study, with SARS-CoV2 infection, but with no association with insufficiency of the 25(OH)D. The patients with infection had 1.38 times likelihood to have lower 25(OH)D levels (respective deficiency), while those with insufficiency had no statistical relevance.

The cardiovascular diseases included in our study were associated mainly with vitamin D deficiency and slightly with insufficient vitamin D, and presented 1.78 times more odds to have 25(OH)D < 20 ng/mL. The role of vitamin D was assessed in coronary artery disease, heart failure and atrial fibrillation and was linked with short-term and long-term prognosis [52,53,54,55]. The influence of vitamin D for cardiovascular disease (CVD) was evaluated in a large study (VITAL) [51] that assessed the role of vitamin D and omega 3 fatty acids as preventive factors for neoplasms and cardiac and vascular pathology [51] with no significant improvement for overall heart diseases mortality and major cardiac events. Vitamin D also showed anticoagulant activity, as it can regulate the expression of procoagulant and antifibrinolytic factors [56].

In this research, vitamin D deficiency was statistically significant associated with metabolic impairment. In our study, we found that obesity relates to vitamin D status, showing a 1.2 times increased risk to have vitamin D insufficiency, but when we addressed dysmetabolic status, we discovered that there was a 1.4 times higher probability to have vitamin D deficiency. Obesity seems to be one of the risk factors for low vitamin D levels [57] and there are more possible explanations: lower variation of vitamin D level through sunlight exposure and seasonal influence in the obese, comparative with normal weight people, impairment of 25-hydroxylation in the liver associated to NAFLD and obesity, larger volume of distribution (with consecutive dilution) and a lessened gene expression for cytochrome P450 in this category of patients [58].

For dyslipidaemia or diabetes mellitus (included in the general frame of metabolic syndrome), it was shown there was an association between lower 25(OH)D levels and poorer glycaemic control in diabetic patients and higher level of cholesterol (LDL fraction) in dyslipidemic patients [59]. Regarding other endocrine disorders, our study showed no particular pattern of vitamin D status, except the fact that patients with thyroid diseases tend to have rather optimal or >20 ng serum 25(OH)D levels, but no higher percentage of deficiency. This could be explained by the awareness and active screening of this patients leading to supplementation (data not available about any supplements of this patients).

The malignancies showed significant association with vitamin D deficiency, patients with malignancies are 2.1 times more likely to have vitamin D deficiency than the patients with no malignant diseases and also had a decreased probability to have sufficient or optimal 25(OH)D levels. In the previous VITAL study, vitamin D was associated with decreased death rates from cancer, but without any significant impact on cancer invasiveness character [51].

Vitamin D has been associated with a series of cancers (prostate, myeloma, colorectal, breast cancer), mediated through the vitamin D receptor (VDR) [60]. Vitamin D has revealed an essential role in the aetiology and management of cancer [60]. Moreover, higher levels of 25(OH)D are present in patients with early stage cancer than in those with advanced/metastatic disease [61,62]. It has been shown that vitamin D may protect against death from cancer [51,62]. The cut-off of vitamin D ≥40 ng/mL has been linked with a low risk for malignancy and all-cause mortality, as a consequences of the pleiotropic effects of the vitamin D [51,63]. A study that evaluated the risk for vitamin D deficiency in women with breast cancer established that 66% of the women had deficit of 25(OH)D at baseline [64].

Although the literature assessed an association between vitamin D and various gastrointestinal diseases, our study showed a feeble relationship between deficient and insufficient 25(OH)D serum values and specific digestive pathology.

The published data indicated an association of vitamin D status with inflammatory bowel disease (IBD) [65,66], non-alcoholic fatty liver disease(NAFLD) [67] and irritable bowel syndrome (IBS) [68,69], but our study showed no specific relevance. Because vitamin D and calcium supplementation in many afflictions was shown to be linked to decreasing of the inflammatory status [70,71], further studies are necessary.

It is stated that mental and behavioural disorders covering a large spectrum are influenced by vitamin D through multiple factors: intrinsic (widespread of 25 (OH) vitamin D receptor in amygdala and neurons, the neuronal calcium regulation, cognitive functions) and extrinsic, such as sun exposure, deficient diet and some medication [72]. In our study, we found an association between vitamin D deficiency and the presence of mental diseases with OR= 1.259.

Regarding the widely studied disorders of musculoskeletal system and vitamin D interrelation, it is well known that lack of vitamin D is involved in the appearance of different degrees of low bone density and sarcopenia [73]. In our study, we found no significant associations between lower levels of vitamin D and diseases of the musculoskeletal system and connective tissue, and the most probable explanation is that those patients are in treatment with vitamin D supplements (we had no access to this data).

Renal impairment could influence the vitamin D level because the process of hydroxylation takes places in the kidneys and the action of 1-alfa hydroxylase is decreased, resulting lower level of active vitamin D [74].

In our research, patients with urinary and renal diseases had 2.3 times more odds to associate vitamin D deficiency, so we can take into consideration to measure the serum level and correct the eventual perturbances or prophylactic supplementation in those patients.

It has also been shown that low levels of 25(OH)D are associated with the development and progression of chronic kidney disease (CKD) and high mortality [75,76,77]. Moreover, vitamin D is essential for protecting kidney function through reducing 24 h urine protein and inflammatory status (CRP, TNF- α, IL-6) in patients with diabetic nephropathy, but without effects on eGFR [75].

The reports from prospective studies on admitted patients described high prevalence of 25(OH) suboptimal in more than >50% of inpatient and an important association with longer hospitalisation, malnourishment, poor quality of life and mortality, with greater rates of deficiency compared with general population, but the relationship of causality was not established [34,78].

In the context of increased awareness of vitamin D supplementation, the use of vitamin D over the counter in high doses revealed the possibility for complications. The literature has showed promising data on this topic showing possible alterations in case of high levels of 25(OH)D described as hypervitaminosis, >250 nmol/L (100 ng/mL) and intoxication, for levels over >375 nmol/L (150 ng/mL), which lead to hypercalciuria and hypercalcemia. The risk for hypervitaminosis from endogenous causes, such as granulomatous disorders or lymphomas has been assessed [30,79,80,81,82]. This study included five patients with vitamin D levels ≥150 ng/mL, but as the main subject of our research was not hypervitaminosis D, serum calcium levels were not evaluated, nor the possible symptomatology of the patients.

To our knowledge, this study is the first and the largest from Romania that reports vitamin D status association with pathology categories.

In this research, the main objective was to assess vitamin D status in a population with a mean of 8–10 h of sunlight (48 N to 43 N) and to establish a correlation with different pathologies. Still our findings must be interpreted with caution because of certain limitations. One of the limitations is the fact that we could not substantiate the relationship of cause and effect and to determine if deficiency of vitamin D firstly contributed to poorer health status or, inversely, the presence of the actual disease led to lower levels of 25(OH)D. We cannot address causality, because 25(OH)D was measured at the time of or after diagnosis, but we can hypothesize that disease-related factors such as reduced outdoor activity due to illness are associated with lower exposure to sunlight, decreased mobility and faulty nutrition and may contribute to 25(OH)D deficiency in hospitalized people. Another limitation of our study is the fact that is difficult to extrapolate our population sample with overall Romanian population and we are unable to compare with general data from healthy people due to no screening recommendations of vitamin D status in the normal population and the existing data are inhomogeneous, although our hospital is a regional one with addressability from all Romanian counties.

The strength of this study is the large contingent of patients, with heterogenous and wide range of pathologies that permitted to analyse multiple subgroups and parameters with high statistical power. Our data are congruent with the literature concerning vitamin D status monitoring in high risk categories, such as hospitalized patients, as it was shown that 25(OH)D deficiency is associated with morbidity in various medical condition subgroups.

## 5. Conclusions

There is an association of suboptimal vitamin D status and pathology, sex and age, especially with cardiovascular disease, malignancy, metabolic diseases, SARS-CoV2 infection, male sex and older age for hospitalised patients. The insufficient 25(OH)D (20–30 ng/mL) stands as the grey zone of vitamin D status, and the monitoring and vitamin D supplementation should be considered for risk categories, and national or international consensus and guidelines are mandatory.

## Figures and Tables

**Figure 1 diagnostics-13-00998-f001:**
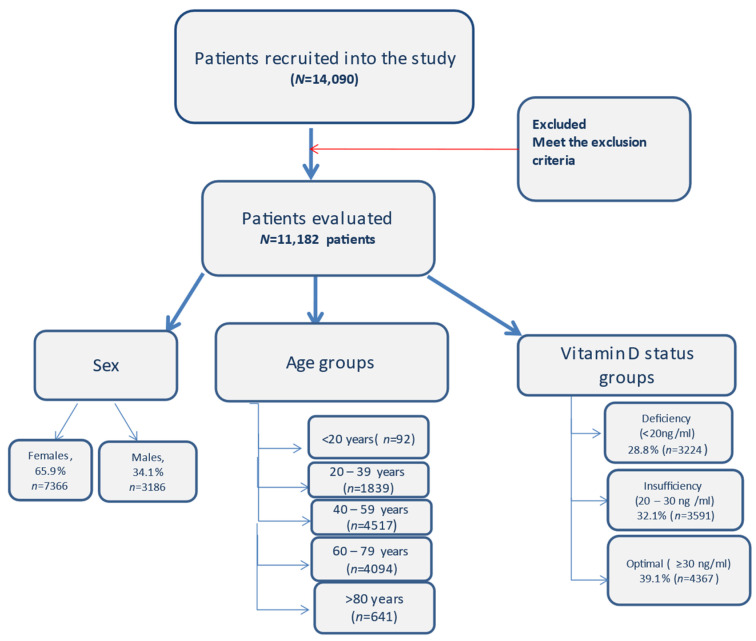
Flow diagram with the patients included in the study and the distribution.

**Figure 2 diagnostics-13-00998-f002:**
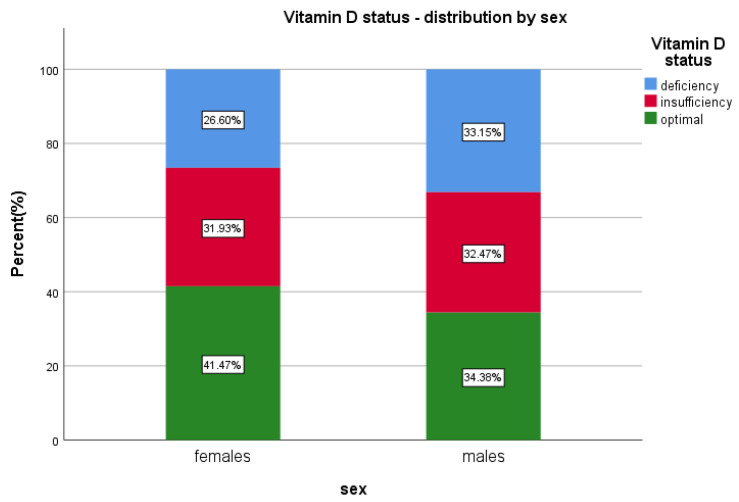
Vitamin D status and sex distribution.

**Figure 3 diagnostics-13-00998-f003:**
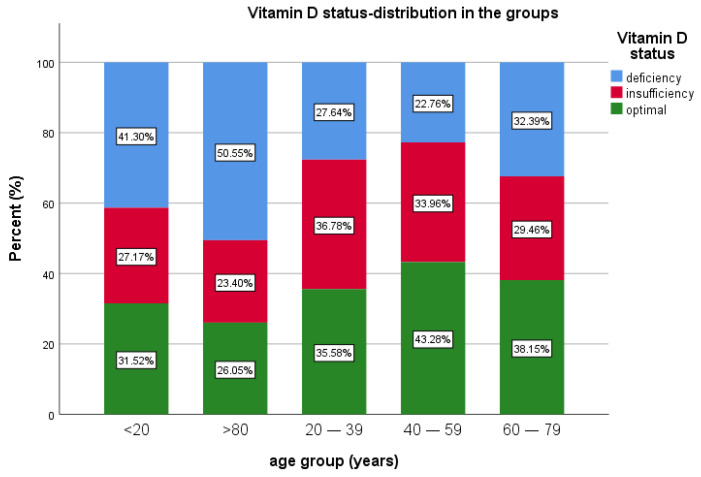
Vitamin D status and age group distribution.

**Table 1 diagnostics-13-00998-t001:** Median (min–max) 25(OH)D and age distribution.

	Overall	Deficiency	Insufficiency	Optimal
Age (median (min–max)) years	55 (18–102)	60 (18–102)	53 (18–95)	54 (18–94)
25(OH)D ng/mL				
Female (median (min–max))	27.2 (3.5–154.2)	15 (3.5–19.9)	25 (20–29.9)	38.3 (30–154.2)
Male (median (min–max))	25.1 (3.5–154.2)	14.2 (3.5–19.9)	24.9 (20–29.9)	37.1 (30–154.2)
Age group (median (min–max))				
<20 years	22.75 (6.5–57.6)	16 (6.5–19.9)	24.2 (20.1–29.1)	38.2 (30.1–57.6)
20–39 years	25.7 (3.5–154.2)	15.7 (3.5–19.9)	24.7 (20–29.9)	37.75 (30–154.2)
40–59 years	27.9 (3.5–154.2)	15.5 (3.5–19.9)	25.1 (20–29.9)	37.5 (30–154.2)
60–79 years	25.85 (3.5–154.2)	14.1 (3.5–19.9)	25.05 (20–29.9)	38.5 (30–154.2)
>80 years	19.9 (3.5–79)	12.75 (3.5–19.9)	25.25 (20–29.9)	38.3 (30–79)
Season (median (min–max))				
spring	24.3 (3.5–154.2)	14.4 (3.5–19.9)	24.6 (20–29.9)	37.9 (30–154.2)
summer	27.5 (3.5–154.2)	15.2 (3.5–19.9)	25.4 (20–29.9)	37.4 (30–154.2)
fall	28.7 (3.5–115.6)	15.45 (3.5–19.9)	25.3 (20–29.9)	37.8 (30–115.6)
winter	25.5 (3.5–154.2)	14.6 (3.5–19.9)	24.6 (20–29.9)	39.2 (30–154.2)

**Table 2 diagnostics-13-00998-t002:** Prevalence of deficiency and sufficiency by sex, age, season, years.

			Optimal (*n* = 4367; 39.05%)		Insufficiency (*n* = 3591; 32.11%)		Deficiency (*n* = 3224; 28.83%)	
		*N* (%)	*N* (%)	*p*-Value	*N* (%)	*p*-Value	*N* (%)	*p*-Value
Sex	Male	3816 (34.12%)	1312 (30.04%)	<0.0001	1239 (34.5%)	0.5634	1265 (39.23%)	<0.0001
	Female	7366 (65.87%)	3055 (69.95%)		2352 (65.5%)		1959 (60.76%)	
Age	<20	92 (0.82%)	29 (0.66%)	<0.0001	25 (0.69%)	<0.0001	38 (1.17%)	<0.0001
	20–39	1838 (16.4%)	654 (14.97%)		676 (18.82%)		508 (15.76%)	
	40–59	4517 (40.4%)	1955 (44.76%)		1534 (42.71%)		1028 (31.96%)	
	60–79	4094 (36.6%)	1562 (35.76%)		1206 (33.58%)		1326 (41.23%)	
	>80	641 (5.7%)	167 (3.82%)		150 (4.17%)		324 (10.07%)	
COVID-19	Yes	1913 (17.11%)	662 (15.15%)	<0.0001	590 (16.43%)	0.1905	661 (20.5%)	<0.0001
	No	9269 (82.89%)	3705 (84.85%)		3001 (83.57%)		2563 (79.5%)	
Season	Winter	2324 (20.8%)	901 (20.6%)	<0.0001	636 (17.7%)	<0.0001	787 (24.4%)	<0.0001
	Spring	3482 (31.1%)	1139 (26.1%)		1105 (30.8%)		1238 (38.4%)	
	Summer	2794 (25.0%)	1147 (26.3%)		982 (27.3%)		665 (20.6%)	
	Fall	2582 (23.1%)	1180 (27.0%)		868 (24.2%)		534 (16.6%)	
Year	2020	2737 (24.5%)	1155 (26.4%)	<0.0001	957 (26.6%)	0.0008	625 (19.4%)	<0.0001
	2021	5606 (50.1%)	2211 (50.6%)		1766 (49.2%)		1629 (50.5%)	
	2022	2839 (25.4%)	1001 (22.9%)		868 (24.2%)		970 (30.1%)	

**Table 3 diagnostics-13-00998-t003:** Prevalence of vitamin D status and pathologies.

Pathology		Optimal (*n* = 4367; 39.05%)		Insufficiency (*n* = 3591; 32.11%)		Deficiency (*n* = 3224; 28.83%)	
First Discharge Diagnostic	*N* (%)	*N* (%)	*p*-Value	*N* (%)	*p*-Value	*N* (%)	*p*-Value
Infectious diseases	115 (1.0%)	24 (0.5%)	<0.0001	22 (0.6%)	0.0007	69 (2.1%)	
Malignant neoplasm	118 (1.1%)	27 (0.6%)		30 (0.8%)		61 (1.9%)	
Benign or uncertain behaviour neoplasms	199 (1.8%)	73 (1.7%)		57 (1.6%)		69 (2.1%)	
Anaemias	80 (0.7%)	26 (0.6%)		25 (0.7%)		29 (0.9%)	
Diseases of blood and blood-forming organs	9 (0.08%)	3 (0.07%)		4 (0.1%)		2 (0.06%)	
Disorders of thyroid gland	2258 (20.2%)	1005 (23.0%)		789 (22.0%)		464 (14.4%)	
Diabetes mellitus	314 (2.8%)	127 (2.9%)		102 (2.8%)		85 (2.6%)	
Disorders of other endocrine glands and pancreas	83 (0.7%)	24 (0.5%)		38 (1.1%)		27(0.7%)	
Malnutrition and nutritional deficiencies	193 (1.7%)	85 (1.9%)		64 (1.8%)		44(1.4%)	
Obesity and other hyperalimentation and metabolic disorders	579 (5.2%)	193 (4.4%)		211 (5.9%)		175 (5.4%)	
Mental and behavioural disorders	507 (4.5%)	166 (3.8%)		171 (4.8%)		170 (5.3%)	
Diseases of the nervous system	407 (3.6%)	178 (4.1%)		117 (3.3%)		112 (3.5%)	
Diseases of the eye and adnexa	5 (0.04%)	4 (0.09%)		1 (0.03%)		0 (0.0%)	
Diseases of the ear and mastoid process	35 (0.3%)	16 (0.4%)		11 (0.3%)		8 (0.2%)	
Acute rheumatic fever and chronic rheumatic heart diseases	2 (0.02%)	1 (0.02%)		0 (0.0%)		1 (0.03%)	
Hypertensive diseases	17 (0.2%)	4 (0.09%)		9 (0.3%)		4 (0.1%)	
Ischaemic heart diseases	135 (1.2%)	52 (1.2%)		47 (1.3%)		36 (1.1%)	
Other forms of heart disease and pulmonary heart diseases	816 (7.3%)	268 (6.1%)		222 (6.2%)		326 (10.1%)	
Diseases of the respiratory system	1552 (13.9%)	542 (12.4%)		497 (13.8%)		513 (15.9%)	
Diseases of oral cavity, salivary glands and jaws	4 (0.04%)	3 (0.07%)		1 (0.03%)		0 (0.0%)	
Diseases of oesophagus, stomach and duodenum	657 (5.9%)	251 (5.7%)		208 (5.8%)		198 (6.1%)	
Hernia and diseases of the appendix	21 (0.2%)	7 (0.2%)		9 (0.3%)		5 (0.2%)	
Noninfective enteritis and colitis	29 (0.3%)	6 (0.1%)		13 (0.4%)		10 (0.3%)	
Other diseases of intestines and diseases of peritoneum	68 (0.6%)	29 (0.7%)		24 (0.7%)		15 (0.5%)	
Liver, billiary and pancreatic diseases	922 (8.2%)	394 (9.0%)		276 (7.7%)		252 (7.8%)	
Other digestive diseases	6 (0.05%)	0 (0.0%)		2 (0.06%)		4 (0.1%)	
Diseases of the skin and subcutaneous tissue	33 (0.3%)	10 (0.2%)		10 (0.3%)		13 (0.4%)	
Diseases of the musculoskeletal system and connective tissue	1419 (12.7%)	638 (14.6%)		454 (12.6%)		327 (10.1%)	
Urinary and renal diseases	217 (1.9%)	55 (1.3%)		57 (1.6%)		105 (3.3%)	
Unspecific	382 (3.4%)	156 (3.6%)		120 (3.3%)		106 (3.3%)	

**Table 4 diagnostics-13-00998-t004:** Distribution of the vitamin D status depending on secondary diagnostic pathologies: cardiovascular disorders, dysmetabolic status, endocrine disorders and malignancy.

			Optimal (*N* = 4367 (39.1%))		Insufficiency (*N* = 3591 (32.1%))		Deficiency (*N* = 3224 (28.8%))	
		*N* (%)	*N* (%)	*p*-Value	*N* (%)	*p*-Value	*N* (%)	*p*-Value
Cardiovascular Disorders	positive	4690 (41.9%)	1602 (36.7%)	<0.0001	1409 (39.2%)	0.0001	1679 (52.1%)	<0.0001
	negative	6492 (58.1%)	2765 (63.3%)		2182 (60.8%)		1545 (47.9%)	
Dysmetabolic Status	positive	4296 (38.4%)	1483 (34.0%)	<0.0001	1386 (38.6%)	0.7906	1427 (44.3%)	<0.0001
	negative	6886 (61.6%)	2884 (66.0%)		2205 (61.4%)		1797 (55.7%)	
Endocrine Disorders	positive	3736 (33.4%)	1464 (33.5%)	0.8389	1201 (33.4%)	0.9583	1071 (33.2%)	0.785
	negative	7446 (66.6%)	2903 (66.5%)		2390 (66.6%)		2153 (66.8%)	
Malignancy	positive	420 (3.8%)	131 (3.0%)	0.0008	99 (2.8%)	0.0001	190 (5.9%)	<0.0001
	negative	10762 (96.2%)	4236 (97.0%)		3492 (97.2%)		3034 (94.1%)	

**Table 5 diagnostics-13-00998-t005:** Group risk and association with vitamin D deficiency.

Association	Risk Group	*p*	OR	95% CI
Vitamin D deficit—Sex	M	<0.0001	1.3687	1.2572 to 1.49
Vitamin D insufficiency—Sex	M	0.5629	1.025	0.9427 to 1.1143
Vitamin D deficit—COVID-19	positive	<0.0001	1.3814	1.2441 to 1.5338
Vitamin D insufficiency—COVID-19	positive	0.189	0.9314	0.8375 to 1.0359
Vitamin D deficit—Cardiovascular disorders	cardiovascular	<0.0001	1.7855	1.6439 to 1.9393
Vitamin D insufficiency—Cardiovascular disorders	cardiovascular	0.0001	0.8483	0.7823 to 0.9198
Vitamin D deficit—Dysmetabolic status	metabolic	<0.0001	1.4086	1.2961 to 1.5308
Vitamin D deficit—Malignancy	malignancy	<0.0001	2.1042	1.7285 to 2.5615
Vitamin D insufficiency—Malignancy	malignancy	*p* = 0.007	0.727	0.577–0.916
Vitamin D deficit—endocrine disorders	endocrine disorders	0.7851	0.988	0.9058 to 1.0776
Vitamin D insufficiency—endocrine disorders	endocrine	0.9583	1.0022	0.9214 to 1.0902

## Data Availability

Datasets generated and/or analysed during the current study are available from the corresponding authors upon reasonable request.

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
