# Peer review of "Association of Vitamin D Deficiency and Insufficiency with Pathology in Hospitalized Patients"

_diagnostics, 2023, doi:10.3390/diagnostics13050998_

Round 1

Reviewer 1 Report

The authors present a study about the ‘Association of vitamin D deficiency and insufficiency with pathology in hospitalized patients. After reading and evaluating the manuscript, I recommend a substantial revision to enhance the quality and understandability of the manuscript.

·       The biomarker for vitamin D status should be mentioned along with the thresholds for vitamin D deficiency and vitamin D insufficiency in accordance with the guidelines for different age groups.

·       Include the statement of novelty. What new has been found or assessed and achieved through this study?

·       I propose including a visual representation of the methodology for improved comprehension and clarity. Also clearly indicated the inclusion and exclusion criteria used in the study.

·       Please specify whether the participants have any history of the

·       The standard error or standard deviation should be indicated in the figures for clarity and reliability.

Author Response

Dear Reviewer,

Thank you for giving us the opportunity to improve our  manuscript “Association of vitamin D deficiency and insufficiency with pathology in hospitalized patients” for publication in Diagnostics  special issue- Vitamin D as a Biomarker in the Standardization Era: A Clinical Point of View.

We appreciate all the time and effort that you dedicated to provide feedback on our manuscript and we are grateful for the insightful comments and highly valuable suggestions that helped us to  improve to our paper.

 We have incorporated your valuable and highly appreciated suggestions in the manuscript, highlighted with track changes:

  1. The biomarker for vitamin D status should be mentioned along with the thresholds for vitamin D deficiency and vitamin D insufficiency in accordance with the guidelines for different age groups.

Response:  According to your well pointed suggestion we mentioned in the text “There are still gaps in establishing thresholds for adequate levels of 25(OH)D and in most recent statements and consensus  it is mentioned that the present data are not sufficient to define certain vitamin D status thresholds, because of the lack of standardization, but at least 20ng/ml 25(OH)D should be achieved, treatment is recommended for values below, and may be considered for serum 25(OH)D<30 ng/ml. Also, the recommendations for dose supplementation or treatment of vitamin D inadequacy vary according to age, risk category, geographical situation and regional authority or  medical society, creating a large heterogeneity in management of vitamin D deficit and leaving place for debates, but there is consistency in agreement that general population screening for vitamin D deficiency is not recommended [14,15,16].

In 2019, Romanian Ministry of Health released a recommendation for vitamin D status assessment in children, pregnant women and adults. According to this, there are  several risk categories of adult population that are exposed to vitamin D deficiency and measurement of 25(OH)D is recommended for adults with: chronic diseases, low ultraviolet light exposure, diseases of liver, renal diseases, maldigestion and malabsorption, cardio-vascular diseases, metabolic and nutritional impairment, cancer, allergies, autoimmune diseases, endocrinopathies and musculoskeletal diseases. [17,18]. Moreover, the pandemic era started in March 2020 and serum 25(OH)D were assessed in the context of COVID-19 risk.

According to Romanian Ministry of Health recommendation for adults, a value of 25(OH)D between 10-20 ng/ml (25-50 nmol/L) is considered deficient, while <10 ng/ml ( <25 nmol/L) represents a severe deficiency.

Although a level >20 ng/ml (>50 nmol/L) may be considered sufficient, it is specified that American Endocrine Society Guidelines suggest that level >30 ng/ml is associated with lower risk of osteomalacia. A level between 20-30 ng/ml is considered insufficiency, while values of 25 OH D> 250 nmol/l (100 ng/ml), are considered toxic [17,1]. Moreover, there are no recommendations of vitamin D status screening in overall healthy population [18].”

  1. Include the statement of novelty. What new has been found or assessed and achieved through this study?

 Response: we stated in our paper that “To our knowledge this study is the first and the largest from Romania that reports vitamin D status association with pathology categories. The strength of this study is the large contingent of patients, with heterogenous and wide range of pathologies that permitted to analyze multiple subgroups and parameters with high statistical power. Our data are congruent with literature concerning vitamin D status monitoring in high risk categories such as hospitalized patients, as it was shown that 25(OH)D deficiency is associated with morbidity in various medical condition subgroups.”

  1. I propose including a visual representation of the methodology for improved comprehension and clarity. Also clearly indicated the inclusion and exclusion criteria used in the study.

 Response: according to your suggestion, we added a visual representation and clearly indicated the inclusion and exclusion criteria in the text

“Inclusion criteria: Patients >18 years old, admitted in Central Military Emergency University Hospital, Bucharest, Romania whom 25(OH)D level were measured in our laboratory and had a discharge diagnostic coded according to International Classification of Diseases, Tenth Revision, Clinical Modification (ICD-10-CM).

Exclusion criteria: patients under 18 years old, outpatients or patients that presented in emergency room, but not admitted in the hospital, patients with multiple admissions, pregnant women, patients with unspecific diagnosis (according to ICD-10-CM).”

  1. Please specify whether the participants have any history of the

Response:  We considered as not being mentioned the history of supplementation and we added in the manuscript “We should consider that in the last few years, vitamin D supplementation was used intensively, especially in the pandemic era and we have no data regarding vitamin D supplementation in our study population.” and “This could be explained by the awareness and active screening of this patients leading to supplementation (data not available about any supplements of this patients).”

  1. The standard error or standard deviation should be indicated in the figures for clarity and reliability.

Response: For more reliability and clarity we added the mentioned data.  We had an abnormal distribution of 25(OH)D values,  so we included table mentioning the median ( min-max) and added data in the manuscript figures.

Sincerely yours,

Sandica Bucurica

Reviewer 2 Report

I've read with attention the paper of Bucurica et al. The intro of the paper is very short and supported by a couple of references only, The method section is clearly reported. The obtained results are reliabe and adequately discussed I have only a couple of concerns:

- The main text and sttistical analyses are simple, but correct.

- The conclusion chapter is long and usually cannot start with the limiation sub-section

- No funding has been repored 

- The text requires an attentive revision before resubmisson

Best regards, 

Author Response

Dear Reviewer,

Thank you for giving us the opportunity to improve our  manuscript “Association of vitamin D deficiency and insufficiency with pathology in hospitalized patients” for publication in Diagnostics  special issue- Vitamin D as a Biomarker in the Standardization Era: A Clinical Point of View

We appreciate the time and effort that you dedicated to provide feedback on our manuscript and we are grateful for the insightful comments and valuable suggestions that helped us to  improve to our paper.

 We have  attentively revised our manuscript and we have incorporated your highly valuable suggestions in the manuscript, highlighted with track changes:

  1. The intro of the paper is very short and supported by a couple of references only, The method section is clearly reported.

Response: According to your highly appreciated suggestion,  we have improved the introduction with relevant information and data to sustain our study, accompanied by pertinent references and we added in the introduction section of our manuscript :

“Vitamin D is a fat soluble vitamin synthetized from precursors as 7-dehydrocholesterol into vitamin D 3 (cholecalciferol) which represents the main source  and from plant ergosterol into vitamin D2, through a multistep process (25-hydroxylation, 1α-hydroxylation, and 24-hydroxylation) to the biologic active form  1,25-dihydroxvitamin D3  [1,25(OH)2D3 ] , respective 1,25(OH)2D 2 [2,3]. The se-rum concentration of 25(OH)D, as an intermediate metabolite is the serum biomarker  of vitamin D status as it represents the main storage form and it is reliable to be measured, because the 1,25(OH)2 D form has around 4 hours half-life, while 25(OH)D reaches 3 weeks.[4]:

 And

“The conventional and well known pathway of vitamin D action involves the steroid vitamin D receptor (VDR) mediation and calcium absorption modulation with major implication in bone metabolism, but there are various non-conventional systemic effects of vitamin D [2].

In cancer, vitamin D and VDR regulation are involved in apoptosis, invasion, inhibition of inflammatory cytokines and regulation of microRNA, but there are equivocal results regarding relation of causality because the of the lack of standardization of 25(OH)D testing method and timing (before and after the diagnosis) [2,7,8,9,107898977].

Regarding the cardiovascular system and vitamin D relationship, it is stated that VDR are found endothelial cells, in smooth muscle tissue and lack of vitamin D promotes atherogenesis [11]. Also VDR is expressed in pancreatic cells and mediates transcription in stellate cells suggesting the mechanism of involvement in metabolic diseases such as diabetes mellitus [12,13].

There are still gaps in establishing thresholds for adequate levels of 25(OH)D and in  most recent statements and consensus  it is mentioned that the present data are not sufficient to define certain vitamin D status thresholds, because of the lack of standardization, but at least 20ng/ml 25(OH)D should be achieved, treatment is recommended for values below, and may be considered for serum 25(OH)D<30 ng/ml.

Also, the recommendations for dose supplementation or treatment of vitamin D inadequacy vary according to age, risk category, geographical situation and regional authority or  medical society, creating a large heterogeneity in management of vitamin D deficit and leaving place for debates, but there is consistency in agreement that general population screening for vitamin D deficiency is not recommended [14,15,16].

In 2019, Romanian Ministry of Health released a recommendation for vitamin D status assessment in children, pregnant women and adults. According to this, there are  several risk categories of adult population that are exposed to vitamin D deficiency and measurement of 25(OH)D is recommended for adults with: chronic diseases, low ultraviolet light exposure, diseases of liver, renal diseases, maldigestion and malabsorption, cardio-vascular diseases, metabolic and nutritional impairment, cancer, allergies, autoimmune diseases, endocrinopathies and musculoskeletal diseases. [17,18]. More-over, the pandemic era started in March 2020 and serum 25(OH)D were assessed in the context of COVID-19 risk. According to Romanian Ministry of Health recommendation for adults, a value of 25(OH)D between 10-20 ng/ml (25-50 nmol/L) is considered deficient, while <10 ng/ml ( <25 nmol/L) represents a severe deficiency.

Although a level >20 ng/ml (>50 nmol/L) may be considered sufficient, it is specified that American Endocrine Society Guidelines suggest that level >30 ng/ml is associated with lower risk of osteomalacia. A level between 20-30 ng/ml is considered insufficiency, while values of 25 OH D> 250 nmol/l (100 ng/ml), are considered toxic [17,119]. Moreover, there are no recommendations of vitamin D status screening in overall healthy population [18].

For inpatients, previous studies showed that  vitamin D deficiency was associated with higher probability of exacerbation of the disease and need for hospitalization [19], with longer hospitalization period and unfavorable outcome and  higher mortality rates in intensive care units [20,21,22]. Also, residents of hospices and hospitalized patients presented the highest risk for deficient vitamin D compared with outpatients, and longer hospitalization of those with lower 25(OH)D and overall hospitalized people tend to have higher rates of deficiency of 25(OH)D [23,24].A particular attention was paid to COVID and vitamin D beginning with the first wave of pandemic, since that multiple observational studies showed that low serum levels of 25(OH)D was associated with severe disease, but not showing a causal relationship. One of the most recent umbrella review of multiple meta-analyses over COVID-19 and vitamin D status reported a significant risk for acquiring the infection, for severity of  progression and for high incidence of the mortality in cases with low serum of 25(OH)D, and it was suggested a potential benefit of vitamin D administrated after the diagnosis [25].Although the etiopathogenetic mechanisms of rickets/osteomalacia is directly related to vitamin D deficiency, the literature is inconsistent regarding the cause and effect relationship between lower 25(OH) D and occurrence or progression of a specific disease or if lack of vitamin D could be a mark of deteriorated health condition.”

  1. The conclusion chapter is long and usually cannot start with the limitation sub-section

Response: Thank you for your useful suggestion that helped us to have  a better manuscript. According to this, we modified the conclusion section to a clearer and shorter one and added limitations and strength in discussion section.

Conclusion: “There is an association of suboptimal vitamin D status and pathology, sex and age, especially with cardiovascular disease, malignancy, metabolic diseases, SARS-CoV2 infection, male sex and older age for hospitalized patients. The insufficient 25(OH)D (20-30 ng/ml) stands as the grey zone of vitamin D status, and the monitoring and vitamin D supplementation should be considered for risk categories, and national or international consensus and guidelines are mandatory.”

Discussion: “To our knowledge this study is the first and the largest from Romania that reports vitamin D status association with pathology categories. One of the limitations is the fact that we could not substantiate the relationship of cause and effect and to determine if deficiency of vitamin D firstly contributed to poorer health status or inversely, the presence of the actual disease led to lower levels of 25(OH)D . We cannot address causality, because 25(OH)D was measured on the time or after diagnosis, but we can hypothesize that disease-related factors like reduced out-door activity due to illness are associated with lower exposure to sunlight,  decreased mobility and faulty nutrition and may contribute to 25(OH)D deficiency in hospitalized people. Another limitation of our study is the fact that is difficult to extrapolate our population sample with overall Romanian population and we are unable to com-pare with general data from healthy people due to no screening recommendations of vitamin D status in normal population and the existing data are inhomogeneous, although our hospital is a regional one with addressability from all Romanian counties.

The strength of this study is the large contingent of patients, with heterogenous and wide range of pathologies that permitted to analyse multiple subgroups and parameters with high statistical power. Our data are congruent with literature concerning vitamin D status monitoring in high risk categories such as hospitalized patients, as it was shown that 25(OH)D deficiency is associated with morbidity in various medical condition subgroups.”

  1. No funding has been reported – we reported in the manuscript the Funding

Sincerely yours,

Sandica Bucurica

Reviewer 3 Report

This manuscript reports a study of a very large group of people with various health problems being treated in a hospital in Romania. The factor being determined was vitamin D status and this was then assessed for statistical significance of association with particular disease conditions. Not surprisingly, a large proportion of subjects had vitamin D status that was either defined as vitamin D insufficiency or vitamin D deficiency. There are two questions raised by this report that were implied but not clearly stated. The first one was: is low vitamin D status contributing to the particular disease processes affecting these subjects? The second question would be: is low vitamin D status a consequence of the disease process as it might limit the exposure of the subjects to the sun in summer or affect the utilisation of vitamin D? As the authors point out, vitamin D insufficiency and deficiency has been reported in different population studies around the world. It is not clear from the findings reported in this study, whether the variation in vitamin D status of the hospitalized patients is similar or different to the variation in vitamin D status of apparently healthy people in the same community and environment. It would be helpful to a reader if it were made clear whether the authors regard their findings as being quite specific to the disease states and quite different from the range of vitamin D status found in the healthy community from which these patients came. It would also be better not to use the expression “serum vitamin D levels” because vitamin D was not being measured in serum. Rather it was 25-hydroxyvitamin D. This difficulty could be overcome by using the term “vitamin D status” instead of “vitamin D levels”.

Author Response

Dear Reviewer,

Thank you for giving us the opportunity to improve our  manuscript “Association of vitamin D deficiency and insufficiency with pathology in hospitalized patients” for publication in Diagnostics  special issue- Vitamin D as a Biomarker in the Standardization Era: A Clinical Point of View

We appreciate the time and effort that you dedicated to provide feedback on our manuscript and we are grateful for the insightful comments  and highly valuable suggestions that helped us to  improve to our paper.

We have incorporated your highly valuable suggestions in the manuscript, highlighted with track changes.

  1. The first one was: is low vitamin D status contributing to the particular disease processes affecting these subjects?

Response: Thank you for your pertinent questions. As the Vitamin D status is still surrounded by controversies, we mentioned the limitations  in  introduction and discussion sections: “For inpatients, previous studies showed that  vitamin D deficiency was associated with higher probability of exacerbation of the disease and need for hospitalization [19], with longer hospitalization period and unfavorable outcome and  higher mortality rates in intensive care units [20,21,22]. Also, residents of hospices and hospitalized patients presented the highest risk for deficient vitamin D compared with outpatients, and longer hospitalization of those with lower 25(OH)D and overall hospitalized people tend to have higher rates of deficiency of 25(OH)D [23,24].

A particular attention was paid to COVID and vitamin D beginning with the first wave of pandemic, since that multiple observational studies showed that low serum levels of 25(OH)D was associated with severe disease, but not showing a causal relationship. One of the most recent umbrella review of multiple meta-analyses over COVID-19 and vitamin D status reported a significant risk for acquiring the infection, for severity of  progression and for high incidence of the mortality in cases with low serum of 25(OH)D, and it was suggested a potential benefit of vitamin D administrated after the diagnosis [25].

Although the etiopathogenetic mechanisms of rickets/osteomalacia is directly related to vitamin D deficiency, the literature is inconsistent regarding the cause and effect relationship between lower 25(OH) D and occurrence or progression of a specific disease or if lack of vitamin D could be a mark of deteriorated health condition.”

  1. The second question would be: is low vitamin D status a consequence of the disease process as it might limit the exposure of the subjects to the sun in summer or affect the utilisation of vitamin D? As the authors point out, vitamin D insufficiency and deficiency has been reported in different population studies around the world.

Response: “. One of the limitations is the fact that we could not substantiate the relationship of cause and effect and to determine if deficiency of vitamin D firstly contributed to poorer health status or inversely, the presence of the actual disease led to lower levels of 25(OH)D . We cannot address causality, because 25(OH)D was measured on the time or after diagnosis, but we can hypothesize that disease-related factors like reduced outdoor activity due to illness are associated with lower exposure to sunlight,  decreased mobility and faulty nutrition and may contribute to 25(OH)D deficiency in hospitalized people. We hypothesise that this could be the result of increasing awareness of vitamin deficiency correction and the possible supplementation due to coronavirus pandemic.”

“The reports from prospective  studies on admitted patients described high prevalence of 25(OH) suboptimal in more than >50% of inpatient and an important association with longer hospitalization, malnourishment, poor quality of life and mortality, with greater rates of deficiency compared with general population, but the relationship of causality was not established [34,78].

  • It is not clear from the findings reported in this study, whether the variation in vitamin D status of the hospitalized patients is similar or different to the variation in vitamin D status of apparently healthy people in the same community and environment. It would be helpful to a reader if it were made clear whether the authors regard their findings as being quite specific to the disease states and quite different from the range of vitamin D status found in the healthy community from which these patients came.

Response : There are no recent data from healthy community from our country, but we found a few and mentioned in our manuscript, because there are no recommendations for screening in general population and we mentioned all this things in our paper.

In introduction section we presented: “In 2019, Romanian Ministry of Health released a recommendation for vitamin D status assessment in children, pregnant women and adults. According to this, there are  several risk categories of adult population that are exposed to vitamin D deficiency and measurement of 25(OH)D is recommended for adults with: chronic diseases, low ultraviolet light exposure, diseases of liver, renal diseases, maldigestion and malabsorption, cardio-vascular diseases, metabolic and nutritional impairment, cancer, allergies, autoimmune diseases, endocrinopathies and musculoskeletal diseases. [17,18]. More-over, the pandemic era started in March 2020 and serum 25(OH)D were assessed in the context of COVID-19 risk.

According to Romanian Ministry of Health recommendation for adults, a value of 25(OH)D between 10-20 ng/ml (25-50 nmol/L) is considered deficient, while <10 ng/ml ( <25 nmol/L) represents a severe deficiency.

Although a level >20 ng/ml (>50 nmol/L) may be considered sufficient, it is specified that American Endocrine Society Guidelines suggest that level >30 ng/ml is associated with lower risk of osteomalacia. A level between 20-30 ng/ml is considered in-sufficiency, while values of 25 OH D> 250 nmol/l (100 ng/ml), are considered toxic [17,119]. Moreover, there are no recommendations of vitamin D status screening in overall healthy population [18].”

Also, in the discussion section we mentioned: ‘’Romanian data from more than 8000 patients showed a high deficiency in elderly patients[32]. Our data are concordant with the literature data sustaining, once again, that people <20 or >60 years old (especially >80) are at risk of having vitamin D deficiency rather than insufficiency (which is more encountered in 40-60 years age group) ,but another study from Romania on patients from private practice and hospital referral with indication for 25(OH)D serum measurement showed a variation of vitamin D deficiency slightly higher for different age groups.[31].

In our study we found relatively similar  prevalence of deficiency compared with literature, slightly lower for 20-39 age groups , 40-59 age group and 60-79 age group, with  mean levels of serum 25(OH)D, at the lower limit especially in 40-59 age group and 60-79 group and lower for 20-39 age group and > 80 years compared with previous published data [31,33].”

“The reports from prospective  studies on admitted patients described high prevalence of 25(OH) suboptimal in more than >50% of inpatient and an important association with longer hospitalization, malnourishment, poor quality of life and mortality, with greater rates of deficiency compared with general population, but the relationship of causality was not established [34,78].”

Also, we found a dynamic of vitamin D status in the general  population over the years from other area studies, since there no relevant reports from Romania and we could relate to Europe in general

“Moreover, French studies on healthy volunteers showed a dynamic of vitamin D status in population over the years, and showed a decreasing tendency of vitamin D  deficiency prevalence from 57.7% to 34.6% in a 2 decades period, but with suboptimal level still prevalent, so updated data are necessary to have an accurate overview and to compare[35].

For Europe there has been reported a prevalence of 53% of suboptimal vitamin D (severe deficiency <12ng/ml prevalence of 13 % and 40% between 12 ng/ml and 29 ng/ml)[16], but there are scarce data from East European area and individual studies showed a poorer vitamin D status (mean 25(OH)levels <20ng/ml) compared with western and northern areas of European continent [14].”

  1. It would also be better not to use the expression “serum vitamin D levels” because vitamin D was not being measured in serum. Rather it was 25-hydroxyvitamin D. This difficulty could be overcome by using the term “vitamin D status” instead of “vitamin D levels”.

Response: Thank you for your attentive observation and according to your great suggestion, we made the changes in the manuscript.

Sincerely yours,

Sandica Bucurica

Reviewer 4 Report

The aim of the study was to evaluate the prevalence of vitamin D deficiency/insufficiency in hospitalised patients, considering demographic factors and to establish a correlation with different comorbidities. There is a strongly statistic significant association between vitamin D deficiency and 405 cardiovascular disease, malignancy, metabolic diseases and SARS-CoV2 infection, comparative with insufficiency or optimal vitamin D . The male sex is a risk factor for vitamin D deficiency, compared with women from our study. The most interesting element is the sample size The introduction is well written , with adequate bibliographic references . However, it should be broader, making reference to previous studies in hospitalized patients. The hypothesis should be considered if it corresponds to an etiopathogenic mechanism or is a marker The methodology is  widely described, which would allow the study to be carried out by another research group. However, the criteria for which vitamin D has been requested or if it is requested for all admitted patients should be indicated Results : The description of the results is  easy to follow and understand. The figures help to understand the results The discussion is correct, adapting to the results obtained. The role of vitamin D as a marker of severe disease should be emphasized in the discussion due to the limitation of mobility that these patients have. The weaknesses and strengths of the study should be included in the discussion, not in the conclusions.    

Author Response

Dear Reviewer,

Thank you for giving us the opportunity to improve our  manuscript “Association of vitamin D deficiency and insufficiency with pathology in hospitalized patients” for publication in Diagnostics  special issue- Vitamin D as a Biomarker in the Standardization Era: A Clinical Point of View

We appreciate the time and effort that you dedicated to provide feedback on our manuscript and we are grateful for the insightful comments  and highly valuable suggestions that helped us to  improve to our paper.

The aim of the study was to evaluate the prevalence of vitamin D deficiency/insufficiency in hospitalized patients, considering demographic factors and to establish a correlation with different comorbidities. There is a strongly statistic significant association between vitamin D deficiency and  cardiovascular disease, malignancy, metabolic diseases and SARS-CoV2 infection, comparative with insufficiency or optimal vitamin D . The male sex is a risk factor for vitamin D deficiency, compared with women from our study. 

Thank your for your appreciation and your great suggestions. We have incorporated your valuable and highly appreciated suggestions in the manuscript, highlighted with track changes:

  1. The most interesting element is the sample sizeThe introduction is well written , with adequate bibliographic references, however, it should be broader, making reference to previous studies in hospitalized patients. The hypothesis should be considered if it corresponds to an etiopathogenic mechanism or is a marker. The methodology is  widely described, which would allow the study to be carried out by another research group.

 Reponse:  We have improved the introduction with relevant information and data to sustain our study, accompanied by pertinent references and we added in the introduction of our manuscript :

“Vitamin D is a fat soluble vitamin synthetized from precursors as 7-dehydrocholesterol into vitamin D 3 (cholecalciferol) which represents the main source  and from plant ergosterol into vitamin D2, through a multistep process (25-hydroxylation, 1α-hydroxylation, and 24-hydroxylation) to the biologic active form  1,25-dihydroxvitamin D3  [1,25(OH)2D3 ] , respective 1,25(OH)2D 2 [2,3]. The se-rum concentration of 25(OH)D, as an intermediate metabolite is the serum biomarker  of vitamin D status as it represents the main storage form and it is reliable to be measured, because the 1,25(OH)2 D form has around 4 hours half-life, while 25(OH)D reaches 3 weeks.[4]:

 And

“The conventional and well known pathway of vitamin D action involves the steroid vitamin D receptor (VDR) mediation and calcium absorption modulation with major implication in bone metabolism, but there are various non-conventional systemic effects of vitamin D [2].

In cancer, vitamin D and VDR regulation are involved in apoptosis, invasion, inhibition of inflammatory cytokines and regulation of microRNA, but there are equivocal results regarding relation of causality because the of the lack of standardization of 25(OH)D testing method and timing (before and after the diagnosis) [2,7-10].

Regarding the cardiovascular system and vitamin D relationship, it is stated that VDR are found endothelial cells, in smooth muscle tissue and lack of vitamin D pro-motes atherogenesis [11]. Also VDR is expressed in pancreatic cells and mediates transcription in stellate cells suggesting the mechanism of involvement in metabolic diseases such as diabetes mellitus [12,13].

There are still gaps in establishing thresholds for adequate levels of 25(OH)D and in  most recent statements and consensus  it is mentioned that the present data are not sufficient to define certain vitamin D status thresholds, because of the lack of standardization, but at least 20ng/ml 25(OH)D should be achieved, treatment is recommended for values below, and may be considered for serum 25(OH)D<30 ng/ml.

Also, the recommendations for dose supplementation or treatment of vitamin D inadequacy vary according to age, risk category, geographical situation and regional authority or  medical society, creating a large heterogeneity in management of vita-min D deficit and leaving place for debates, but there is consistency in agreement that general population screening for vitamin D deficiency is not recommended [14-16].

For inpatients, previous studies showed that  vitamin D deficiency was associated with higher probability of exacerbation of the disease and need for hospitalization [19], with longer hospitalization period and unfavorable outcome and  higher mortality rates in intensive care units [20,21,22]. Also, residents of hospices and hospitalized patients presented the highest risk for deficient vitamin D compared with outpatients, and longer hospitalization of those with lower 25(OH)D and overall hospitalized people tend to have higher rates of deficiency of 25(OH)D [23,24].A particular attention was paid to COVID and vitamin D beginning with the first wave of pandemic, since that multiple observational studies showed that low serum levels of 25(OH)D was associated with severe disease, but not showing a causal relationship. One of the most recent umbrella review of multiple meta-analyses over COVID-19 and vitamin D status reported a significant risk for acquiring the infection, for severity of  progression and for high incidence of the mortality in cases with low serum of 25(OH)D, and it was suggested a potential benefit of vitamin D administrated after the diagnosis [25].Although the etiopathogenetic mechanisms of rickets/osteomalacia is directly related to vitamin D deficiency, the literature is inconsistent regarding the cause and effect relationship between lower 25(OH) D and occurrence or progression of a specific disease or if lack of vitamin D could be a mark of deteriorated health condition.”

  1. However, the criteria for which vitamin D has been requested or if it is requested for all admitted patients should be indicated 

Response: We added in the Introduction section the criteria for vitamin D measurement  in our cohort

“In 2019, Romanian Ministry of Health released a recommendation for vitamin D status assessment in children, pregnant women and adults. According to this, there are  several risk categories of adult population that are exposed to vitamin D deficiency and measurement of 25(OH)D is recommended for adults with: chronic diseases, low ultraviolet  light exposure, diseases of liver, renal diseases, maldigestion and malabsorption,  cardio-vascular diseases, metabolic and nutritional impairment, cancer, allergies, autoimmune diseases, endocrinopathies and musculoskeletal diseases. [17,18]. More-over, the pandemic era started in March 2020 and serum 25(OH)D were assessed in the context of COVID-19 risk.  According to Romanian Ministry of Health recommendation for adults, a value of 25(OH)D between 10-20 ng/ml (25-50 nmol/L) is considered deficient, while <10 ng/ml ( <25 nmol/L) represents a severe deficiency.

Although a level >20 ng/ml (>50 nmol/L) may be considered sufficient, it is specified  that American Endocrine Society Guidelines suggest that level >30 ng/ml is associated  with lower risk of osteomalacia. A level between 20-30 ng/ml is considered insufficiency, while values of 25 OH D> 250 nmol/l (100 ng/ml), are considered toxic [17,119]. Moreover, there are no recommendations of vitamin D status screening in overall healthy population [18].”

  1. The role of vitamin D as a marker of severe disease should be emphasized in the discussion due to the limitation of mobility that these patients have.

Response: According to your highly appreciated suggestion we emphasized in the discussion section: “One of the limitations is the fact that we could not substantiate the relationship of cause and effect and to determine if deficiency of vitamin D firstly contributed to poorer health status or inversely, the presence of the actual disease led to lower levels of 25(OH)D . We cannot address causality, because 25(OH)D was measured on the time or after diagnosis, but we can hypothesize that disease-related factors like reduced outdoor activity due to illness are associated with lower exposure to sunlight,  decreased mobility and faulty nutrition and may contribute to 25(OH)D deficiency in hospitalized people.”

  1. The weaknesses and strengths of the study should be included in the discussion, not in the conclusions.    

Response: Thank you for your useful suggestion that helped us to have  a better manuscript .

We included the strength and limits of our study in the discussion section :

“To our knowledge this study is the first and the largest from Romania that reports vitamin D status association with pathology categories. One of the limitations is the fact that we could not substantiate the relationship of cause and effect and to determine if deficiency of vitamin D firstly contributed to poorer health status or inversely, the presence of the actual disease led to lower levels of 25(OH)D . We cannot address causality, because 25(OH)D was measured on the time or after diagnosis, but we can hypothesize that disease-related factors like reduced out-door activity due to illness are associated with lower exposure to sunlight,  decreased mobility and faulty nutrition and may contribute to 25(OH)D deficiency in hospitalized people. Another limitation of our study is the fact that is difficult to extrapolate our population sample with overall Romanian population and we are unable to com-pare with general data from healthy people due to no screening recommendations of vitamin D status in normal population and the existing data are inhomogeneous, although  our hospital is a regional one with addressability from all Romanian counties.

The strength of this study is the large contingent of patients, with heterogenous and wide range of pathologies that permitted to analyze multiple subgroups and parameters  with high statistical power. Our data are congruent with literature concerning vitamin D status monitoring in high risk categories such as hospitalized patients, as it was shown that 25(OH)D deficiency is associated with morbidity in various medical condition subgroups.”

 Sincerely yours,

Sandica Bucurica

Round 2

Reviewer 1 Report

I have read and evaluated the revised manuscript, and I recommend acceptance.

Reviewer 3 Report

No comments to authors

Reviewer 4 Report

The questions have been answered by the authors